# Effect of Coffee Berry Extract on Anti-Aging for Skin and Hair—In Vitro Approach

Nisakorn Saewan [1,2]

1 School of Cosmetic Science, Mae Fah Luang University, 333, Moo.1, Thasud, Muang, Chiang Rai 57100, Thailand; nisakorn@mfu.ac.th
2 Cosmetic and Beauty Innovations for Sustainable Development (CBIS) Research Group, Mae Fah Luang University, 333, Moo.1, Thasud, Muang, Chiang Rai 57100, Thailand

**Abstract:** The aging process encompasses gradual and continuous changes at the cellular level that slowly accumulate with age. The signs of aging include many physiological changes in both skin and hair such as fine lines, wrinkles, age spots, hair thinning and hair loss. The aim of the current study was to investigate the anti-aging potential of coffee berry extract (CBE) on human dermal fibroblast (HDF) and hair follicle dermal papilla (HFDP) cells. Coffee berry was extracted by 50% ethanol and determined for chemical constituents by HPLC technique. Cytotoxicity of the extract was examined on both cells by MTT assay. Then, HDF cells were used to evaluate antioxidant properties by using superoxide dismutase activity (SOD) and nitric oxide inhibition as well as anti-collagenase inhibition assays. The effectiveness of anti-hair loss properties was investigated in HFDP cells by considering cell proliferation, 5α-reductase inhibition (5AR), and growth factor expression. The results showed that caffeine and chlorogenic acid were identified as major constituents in CBE. CBE had lower toxicity and cell proliferation than caffeine and chlorogenic acid on both cells. CBE showed SOD and nitric oxide inhibition activities that were higher than those of caffeine but lower than those of chlorogenic acid. Interestingly, CBE had the highest significant anti-collagenase activity, and its 5AR inhibition activity was comparable to that of chlorogenic acid, which was higher than caffeine. CBE also stimulated hair-related gene expression, especially insulin-like growth factor 1 (IGF-1), keratinocyte growth factor (KGF) and vascular endothelial growth factor (VEGF). The results confirmed that CBE provided anti-aging activity on both skin and hair cells and could be beneficial for applications in cosmeceuticals.

**Keywords:** anti-aging; antioxidant; growth factor; hair growth; hair loss; superoxide dismutase; 5α-reductase inhibition

## 1. Introduction

Aging is a biological process defined as a gradual and continuous decline over time in cellular and organismal function during a lifetime that ultimately leads to senescence. Skin aging occurs as the result of intrinsic aging (natural consequence and genetics) and extrinsic aging (skin's response to external damage). The signs of skin aging include hyperpigmentation, loss of elasticity and laxity, fine lines and wrinkles, telangiectasia, uneven texture, enlarged pores, baggy eyes, keratosis, etc. [1]. Hair aging comprises weathering of the hair shaft and aging of the hair follicle, which shows striking changes in hair diameter, luster, texture, and loss [2]. To prevent and attenuate the aging process, many research studies worldwide has been conducted in searching for active ingredients with ability to restore skin elasticity, slow the formation of skin wrinkles, and reduce hair thinning and loss. Plant extracts are popularly used in both skin and hair anti-aging cosmetics because their wide variety of functions, including those of antioxidants and immunostimulants, and for radical-scavenging, UV protection, 5-AR inhibition, hair growth promotion, etc. [1,2].

Coffee, a popular beverage worldwide, is made from the roasted coffee bean. Regularly drinking coffee has been reported to reduce the oxidation of human low-density lipoprotein (LDL) and the risk of atherosclerosis as well as improve psychoactive responses, neurological conditions and metabolic disorders [3,4]. Coffee berry is the fruit of coffee plants, which when ripe is round with red and purple colors. In general, each coffee berry contains 2 seeds (beans) that are usually green in color and known as green bean or green coffee. The bean is mainly endosperm tissue surrounded by the endocarp and testa (silverskin) [5]. In the processing of coffee, the husk and pulp are removed as waste byproducts from the coffee beans, which are further roasted to obtain coffee aroma.

Coffee is a rich source of antioxidant and prooxidant compounds such as caffeine and chlorogenic acids [6]. Recently, there has been an attempt to expand the uses of coffee in personal care products due to its antioxidant and anti-aging properties. Caffeine is an antioxidant that neutralizes excess free radicals, inhibits lipid peroxidation, protects cells from free radical damage, improves cell oxygenation and microcirculation and stimulates cell metabolism [7,8]. Chlorogenic acid is an important biologically active polyphenol that has important antioxidant, anti-inflammatory, and anti-microbial properties [9–11]. Furthermore, chlorogenic acid significantly reduces the wound area size during the inflammatory phase [12]. Desai and Mallya reported that green coffee bean extract showed anti-elastase activity in L929 cell lines without toxicity [13]. Zofia et al. demonstrated that green coffee bean and kombucha extracts exhibit anti-aging properties through inhibition of enzyme collagenase and elastase activities in keratinocyte and fibroblast cell lines [14]. Coffee silverskin extracts have the potential to reduce the production of intracellular ROS in keratinocytes, improve hydration and firmness, and protect against skin photoaging induced by UV radiation [15]. In our previous study, extract of coffee byproducts showed high phenolic content with good antioxidant activities in Ferric reducing power, DPPH radical scavenging, nitric oxide inhibition, and increased superoxide dismutase activity [16]. The literature has demonstrated that all parts of coffee fruit contain bioactive compounds. Using the whole fruit has high potential for application in new anti-aging formulations. In a study by McDaniel, twice-daily application of 1% coffee berry extract in cream and 0.1% extract in cleanser showed improvement of skin appearance, fine lines and wrinkles, roughness and dryness, and skin pigmentation. Skin biopsies showed that application of the products reduced MMP-1 and IL-1β. The extract also showed up-regulation of gene expression for four collagen structural proteins and down-regulated gene expression for three MMPs [17].

Although various biological activities of coffee extracts have been reported and cosmetics containing coffee extract are available in the market [16,18–20], information about the mechanism of action of coffee berry in human cell-based assays, in comparison with its major compounds, caffeine and chlorogenic acid, is limited. To obtain useful information for cosmetic applications, this study aimed to investigate skin anti-aging and anti-hair loss along with hair growth promoting potential of CBE and its major components in human skin (HDF) and hair (HFDP) cells. HDF cells were used to determine skin antioxidant, cell proliferation and anti-collagenase activities. Additionally, HFDP cells were applied to evaluate the effect of the extract on hair cell proliferation, 5α-reductase inhibition, and gene expression of IGF-1, KGF, hepatocyte growth factor (HGF), and VEGF. In all experiments, caffeine and chlorogenic acid, the predominant compounds in coffee extract, were used as positive compounds.

## 2. Materials and Methods

Ethanol, sulfuric acid, and trichloroacetic acid were purchased from Merck, Darmstadt, Germany. 1,1-Diphenyl-2-picrylhydrazyl (DPPH), caffeine, chlorogenic acid, folin-ciocalteu reagent, gallic acid, lipopolysaccharide (LPS), phenylmethanesulfonyl fluoride, sodium carbonate, Tris/HCl, triton x-100, vanillin and SOD assay Kit-WST were purchased from Sigma-Aldrich Co., St. Louis, MO, USA. Ferric chloride and potassium ferricyanide were purchased from Fisher scientific, Waltham, MA, USA.

Dimethyl sulfoxide (DMSO) and 3-(4,5-dimethylthiazol-2-yl)-2,5-diphenyltetrazolium bromide (MTT) were purchased from Bio Basic Inc., Markham, ON, Canada. Penicillin streptomycin solution, phosphate buffered saline (PBS), and trypsin/EDTA solution were purchased from Gibco, Grand Island, NY, USA. Fetal bovine serum, human hair follicle dermal papilla cells (C-12071), follicle dermal papilla cell growth medium (C26501), human dermal fibroblast (C-12302) and fibroblast growth medium (C-23020) were purchased from PromoCell GmbH, Heidelberg, Germany.

### 2.1. Extraction

The fresh coffee berries were collected from Chiang Rai, Thailand in September 2019. The berries were washed and air dried at ambient temperature. Then, they were crushed into small pieces using a blender. The coffee berries (100 g) were soaked in 50% ethanol (500 mL) in a sonication bath at 20 MHz at room temperature for 30 min. The extract was filtered with Whatman No. 1 filter paper. Then, the extracts were evaporated by a rotary evaporator (Buchi R-114 Rotary Vap System) at 40 °C and stored at 4 °C until being analyzed.

### 2.2. Determination of Caffeine and Chlorogenic Acid

The quantities of caffeine and chlorogenic acid were analyzed using high-performance liquid chromatography (HPLC, 1290 Infinity II LC System, Agilent Technologies, Waldbronn, Germany) with photodiode array (PDA) detection at 280 nm. CBE was filtered through Whatman® membrane filters with nylon pore size of 0.2 μm, and then, 10 μL of the extract was injected into a Poroshell 120 EC-C18 (250 mm o× 4.6 mm, 4 μm). The mobile phase systems were (A) acetonitrile and 1.5% acetic acid with a ratio of 10:90, and (B) acetonitrile and 1.5% of acetic acid with a ratio of 15:85; elution with a flow rate of 1 mL/min of A from 0 to 6 min and B from 6 to 12 min. The retention times of chlorogenic acid and caffeine standards were 7.21 and 7.86 min. The peak areas were automatically integrated, and the chromatograms were plotted and processed by computer using Agilent software. The standard calibration curves were created by plotting peak areas obtained from HPLC analysis against concentrations of standard solution. The stock solutions of caffeic acid and chlorogenic acid were dissolved in DMSO and diluted to give concentrations of 5, 10, 25, 50 and 100 μg/ mL for preparation of the calibration range. The calibration curves of both standards were fitted by linear regression. The concentrations of caffeine and chlorogenic acid were calculated using the calibration curve of the standards.

### 2.3. Cell Culture

HDF and HFDP cells were cultured in medium supplemented with 10% fetal bovine serum and 1% penicillin streptomycin solution. Cells were cultured until reaching 90% confluence, and then, cells (20,000 cells/mL) were counted into new microwell plates and incubated at 37 °C in a 5% $CO_2$ humidified incubator for 24 h before being used for further determination.

### 2.4. Cytotoxicity

HDF and HFDP cells were used to investigate the cytotoxicity of the test samples by MTT assay [21]. Both cells were treated with the diluted extract, caffeine and chlorogenic acid at a concentration of 0–5 mg/mL for 24 h. Afterward, the culture medium was removed, and 50 μL of MTT solution (0.1 mg/mL) was added to each well and incubated for a further 4 h. DMSO (100 μL) was added to all wells and incubated at room temperature for 30 min. The absorbance of each well was measured at 570 nm using a microplate reader (Biochrom, Cambridge, UK). The percentage of cell viability was calculated using the formula:

$$\text{Viable cell (\%)} = \frac{(A_{\text{treated group}})}{(A_{\text{untreated group}})} \times 100$$

The cytotoxicity of test samples was expressed in terms of $IC_{50}$, which is the concentration of test sample needed to induce 50% inhibition of cell growth, as compared with untreated control cells. Then, the non-cytotoxic concentration of the sample was chosen to use as the test concentration in further experiments.

### 2.5. Cell Proliferation

HDF and HFDP cell proliferation-promoting activity was determined by a modified MTT proliferation method [22]. The samples with the maximum nontoxic dose were added to cells (20,000 cells/mL) and incubated at 37 °C in a 5% $CO_2$ humidified incubator for 72 h. Then, the culture medium was removed. A 50 µL amount of filtered sterilized MTT solution (0.1 mg/mL) was added and incubated for 4 h. At the end of the incubation, 100 µL of dimethylsulfoxide was added, followed by incubatation for a further 30 min. The absorbance was measured at 570 nm using a microplate reader (Biochrom, Holliston, MA, USA). The proliferation of both cells was calculated according to the following equation:

$$\text{Cell proliferation (\%)} = \frac{(A_{sample} - A_{control})}{(A_{control})} \times 100$$

where $A_{control}$ is the absorbance of the control (without test sample), and $A_{sample}$ is the absorbance in the presence of the test sample.

### 2.6. Antioxidant Activity by Superoxide Dismutase

Superoxide dismutase activity on cells was determined by using the modified SOD determination kit [22]. HDF cells were supplemented with 100 µL of the sample and incubated for 24 h. Then, cells were harvested with 0.05% trypsin solution. Cells were lysed with lysis buffer (20 mM Tris/HCl, 0.5 mM phenylmethanesulfonyl and 0.2% triton x-100) and centrifuged at 3000 rpm for 15 min. The SOD activity in supernatants was measured by using SOD assay Kit (Sigma-Aldrich, St. Louis, MO, USA). Briefly, 20 µL of supernatant was added with 200 µL of WST working solution and 20 µL of enzyme working solution. Then, the mixture was incubated at 37 °C for 20 min, and the absorbance measured at 450 nm using a microplate reader (Biochrom, Holliston, MA, USA). The SOD activity was calculated using the following equation:

$$\text{SOD activity (\%)} = \frac{(A_{control} - A_{sample})}{(A_{control})} \times 100$$

where $A_{control}$ is the absorbance of the control (without extract), and $A_{sample}$ is the absorbance in the presence of the test sample.

### 2.7. Antioxidant Activity by Nitric Oxide Inhibition

HDF cells were supplemented with 100 µL of sample before being stimulated with 1 µg/mL lipopolysaccharide (LPS) and incubated for 24 h. The nitric oxide production was assessed using the Griess reagent system (Promega, Madison, WI, USA). In brief, the culture medium (50 µL) was combined with 50 µL of sulfanilamide solution and 50 µL of 0.1% N-1-napthylethylenediamine dihydrochloride solution. Then, the mixture was incubated at room temperature for 5 min, and the absorbance was measured at 540 nm using a microplate reader (Biochrom, Holliston, MA, USA). The amount of nitrite in the samples was determined using a sodium nitrite standard curve, and the percentage of nitric oxide inhibition was calculated as follows:

$$\text{Inhibition of nitric oxide (\%)} = \frac{(A_{control} - A_{sample})}{(A_{control})} \times 100$$

where $A_{control}$ is the absorbance of the control (without extract), and $A_{sample}$ is the absorbance in the presence of the test sample.

## 2.8. Anti-Collagenase Activity

Matrix metalloproteinase-1 (MMP-1) colorimetric drug discovery kit, which was designed to screen MMP-1 inhibitors using a thiopeptide as a chromogenic substrate, was used to determine collagenase inhibition activity. HDF cells were supplemented with a 100 μL sample and incubated for 24 h. Then, cells were harvested with 0.05% trypsin solution. Cells were lysed with lysis buffer and centrifuged at 3000 rpm for 15 min. The supernatants, 20 μL of 153 mU/μL MMP-1 and 20 μL of 1.3 μM prototypic control inhibitor, were mixed and incubated at 37 °C for 60 min to allow interaction between inhibitor and enzyme. Next, 10 μL of 100 μM thiopeptide was added, and then, the absorbance was measured at 412 nm. The percentage of collagenase inhibition was calculated as:

$$\text{Collagenase inhibition (\%)} = \frac{(A_{\text{control}} - A_{\text{sample}})}{(A_{\text{control}})} \times 100$$

where $A_{\text{control}}$ is the absorbance of the control (without test sample), and $A_{\text{sample}}$ is the absorbance in the presence of the test sample.

## 2.9. 5α-Reductase Inhibition

HFDP cells were cultured with medium without fetal bovine serum for 24 h. Then, the cells were incubated with the samples for 24 h. Total *RNAs* were extracted by acid guanidinium thiocyanate-phenol-chloroform method. Then, 2 mg of total *RNAs* were reverse transcribed with M-MLV reverse transcriptase in the presence of random hexamer. The cDNA obtained from this reaction was used as a DNA template for PCR reactions. The PCR-thermal profile started with an initial denaturation at 94 °C for 3 min, followed by 35 cycles of denaturation at 94 °C for 30 s, annealing at 52 °C for 30 s, and extension at 72 °C for 2 min, followed by a final extension at 72 °C for 10 min. The PCR products were analyzed using 1% agarose gel electrophoresis and quantified using a densitometer [23].

## 2.10. Growth Factor Gene Expression

HFDP cells were cultured with medium without fetal bovine serum for 24 h. Then, the cells were incubated with the samples for 24 h. Total *RNAs* were extracted by acid guanidinium thiocyanate-phenol-chloroform method. Two milligrams of total *RNAs* were reverse transcribed with M-MLV reverse transcriptase in the presence of random hexamer. The resultant RT mixtures were then subjected to PCR cycles as follows: 94 °C for 30 s, 58 °C for 30 s, 72 °C for 1 min for 40 cycles (IGF-1, HGF and VEGF) and 35 cycles (KGF) [24]. The nucleotide sequences of the primers and expected sizes of PCR products are shown in Table 1.

**Table 1.** Nucleotide sequences of primers and expected sizes of PCR products.

| Growth Factor | | Primer Sequence | Expected Size (bp) |
|---|---|---|---|
| IGF-1 | Forward (5′–3′) | TCAACAAGCCCACAGGGTAT | 307 |
| | Reverse (5′–3′) | ACTCGTGCAGAGCAAAGGAT | |
| HGF | Forward (5′–3′) | CGAGGCCATGGTGCTATACT | 297 |
| | Reverse (5′–3′) | ACACCAGGGTGATTCAGACC | |
| KGF | Forward (5′–3′) | GACATGGATCCTGCCAACTT | 304 |
| | Reverse (5′–3′) | AATTCCAACTGCCACTGTCC | |
| VEGF | Forward (5′–3′) | TCTTCAAGCCATCCTGTGTG | 297 |
| | Reverse (5′–3′) | GCGAGTCTGTGTTTTTGCAG | |

## 2.11. Statistical Analysis

All measurements were performed in triplicate. The obtained data were statistically analyzed using SPSS 11.5 for Windows (SPSS Inc., Chicago, IL, USA), and the differences were considered significant when $p < 0.05$. Data comparison was analyzed by using one way analysis of variance (ANOVA) with Duncan's multiple range test.

## 3. Results and Discussion

### 3.1. Determination of Caffeine and Chlorogenic Acid

Coffee contains a large number of chemical compounds such as alkaloids (caffeine, trigonelline), phenolics (chlorogenic acid (as main phenolic), epicatechin, catechin, rutin, protocatechuic acid and ferulic acid) and polymers (melanoidins) [25]. The well-known major bioactive constituents of coffee are caffeine and chlorogenic acid [26]. Caffeine increases blood circulation, promotes healthier hair follicles, and reduces hair loss by inhibiting 5α-reductase. Therefore, it has been a popular component of commercial anti-hair loss products. Chlorogenic acid exhibits antioxidant, anti-inflammatory, anti-microbial, and wound healing properties that are related to skin anti-aging. In this study, coffee berries were extracted with 50% ethanol, and the presence of caffeine and chlorogenic acid was determined by HPLC analysis. The chromatogram of CBE at a UV detection wavelength of 280 nm showed two major peak signals that were identified as chlorogenic acid and caffeine at the retention time of 7.21 and 7.86 min, respectively (Figure 1). The concentrations of chlorogenic acid and caffeine were calculated from the peak area against the standard calibration curve and showed values of $34.82 \pm 0.79$ and $22.29 \pm 2.73$ µg/mL extract, respectively.

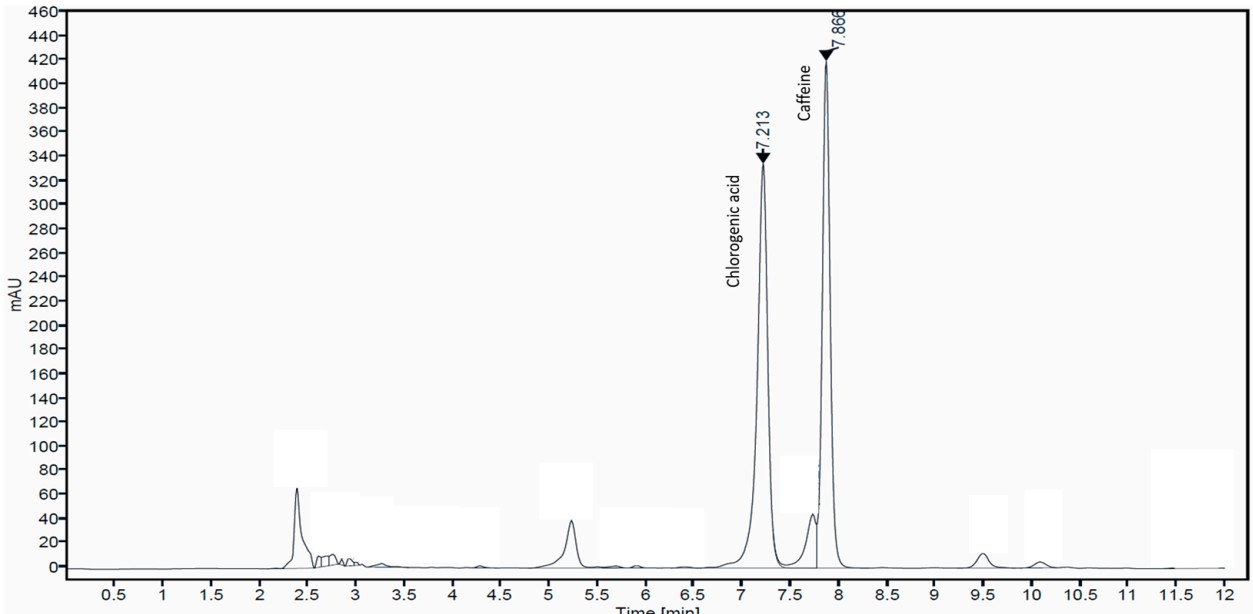

**Figure 1.** HPLC chromatogram of CBE.

### 3.2. Cytotoxicity

First, the cytotoxicity of CBE, caffeine and chlorogenic acid was evaluated by treating HDF and HFDP cells with various concentrations (0–5 mg/mL) of the samples. All test samples showed cytotoxicity in a dose-dependent manner and had a similar pattern. At the low concentration, CBE ($\leq 1.2$ mg/mL), caffeine ($\leq 0.2$ mg/mL) and chlorogenic acid ($\leq 0.2$ mg/mL) displayed non-cytotoxicity to both cells and also stimulated a small percentage of cell proliferation, and, as expected, the increase in the concentration decreased the cells' viability (Figure 2A,B). The test sample concentration with cell viability of 50%, $IC_{50}$, was calculated from dose–response data as compared with control (untreated cells) data. The $IC_{50}$ values for all components on both cell types are summarized in Table 2. In comparison to HDF cells, chlorogenic acid and caffeine showed slightly higher toxicity on HFDP cells with lower $IC_{50}$, while CBE showed no difference in effect. The significantly higher $IC_{50}$ of the extract (3.08 and 3.07 mg/mL for HDF and HFDP, respectively) indicated a lower level of toxicity than that of chlorogenic acid (2.71 and 2.62 mg/mL for HDF and HFDP, respectively) and caffeine (2.63 and 2.38 mg/mL for HDF and HFDP, respectively). It can be presumed that the lower toxicity of CBE is due to the contribution of other

compounds present in this particular extract. Other studies have also observed that coffee silver skin extract had potential antidiabetic property without toxicity in INS-1E cells [27]. Ethanol extract of coffee cherry at a concentration of 10 mg/mL did not show any irritation in an HET-CAM assay [28]. Abd et al. reported that 3.6% caffeine did not induced any toxicity in a normal human keratinocyte cell line. The results suggested that the coffee extract is considered safe and non-toxic for human skin [29]. Hence, the maximum non-toxic doses of CBE (1.2 mg/mL), caffeine (0.2 mg/mL), and chlorogenic acid (1.2 mg/mL) were chosen to use as the test concentrations in further experiments.

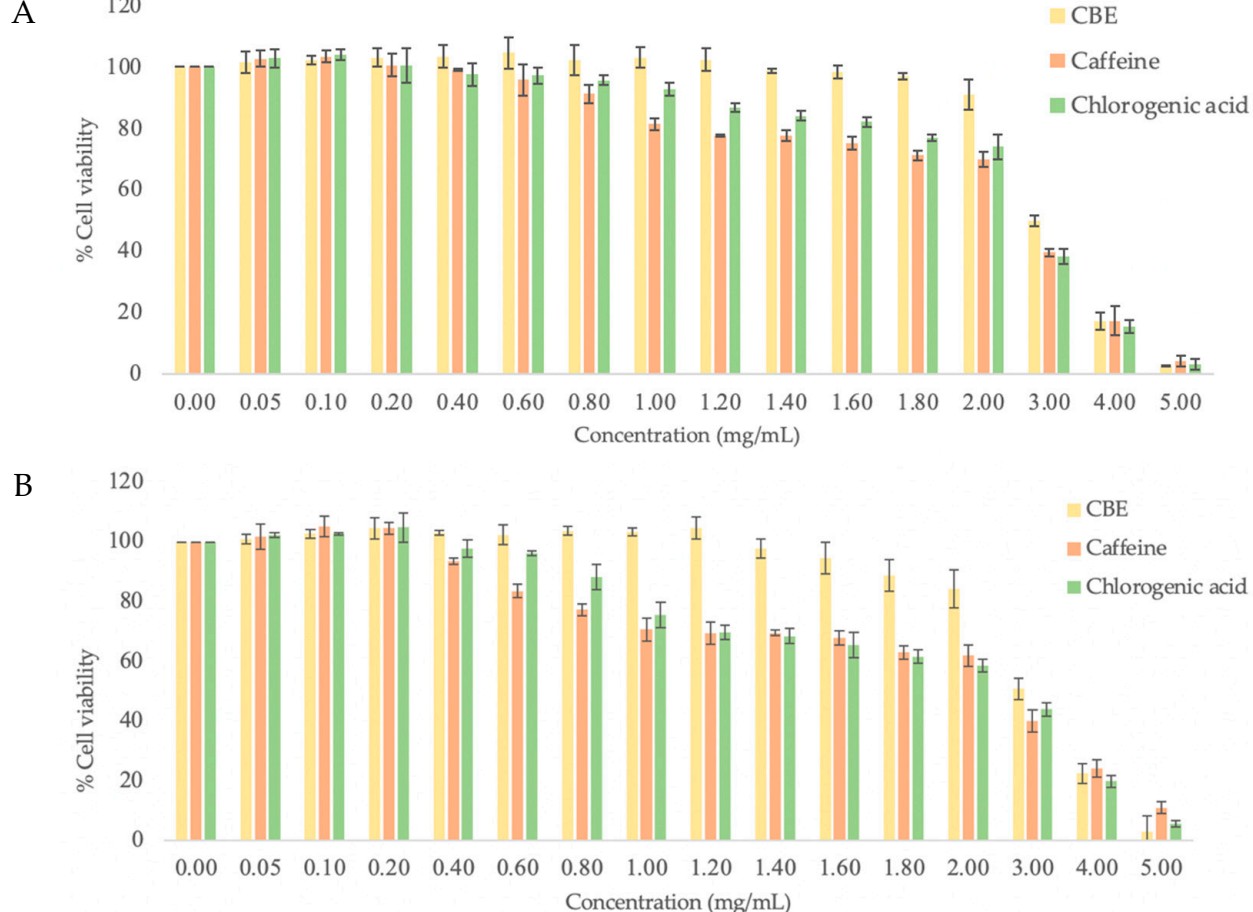

**Figure 2.** Cytotoxicity of various concentrations of CBE, caffeine, and chlorogenic acid on (**A**) HDF and (**B**) HFDP cells.

**Table 2.** The maximum non-toxic dose and 50% inhibition concentration ($IC_{50}$) of CBE, caffeine, and chlorogenic acid on HDF and HFDP cells.

| Samples | HDF | | HFDP | |
| --- | --- | --- | --- | --- |
| | Maximum Non-Toxic Dose (mg/mL) | $IC_{50}$ (mg/mL) | Maximum Non-Toxic Dose (mg/mL) | $IC_{50}$ (mg/mL) |
| CBE | ≤1.2 | 3.08 ± 0.04 [a] | ≤1.2 | 3.07 ± 0.09 [a] |
| Caffeine | ≤0.2 | 2.63 ± 0.14 [b] | ≤0.2 | 2.38 ± 0.12 [b] |
| Chlorogenic acid | ≤0.2 | 2.71 ± 0.03 [b] | ≤0.2 | 2.62 ± 0.04 [b] |

Data with different lower-case letters (a and b) indicate significant differences ($p < 0.05$) between samples.

### 3.3. Skin Anti-Aging Effect

Skin is the largest organ of the human body and is susceptible to the aging process due to exposure to the harsh external environment [30]. Skin aging is the result of a loss of cellular function, gradual loss of the homeostatic mechanism, and disruption in

the structure of skin tissues [31]. These mechanisms are directly linked to skin aging phenotypes: wrinkle formation, uneven pigmentation and decreased wound healing [32]. Recently, many alternative cell-based assays have been used to investigate the cosmetic potential of compounds instead of painful in vivo animal tests as a way to reduce or eliminate harm to animals. Human skin cells, especially fibroblast cells, are used to investigate antioxidant, anti-inflammatory, wound healing, collagen synthesis and cell proliferation properties of natural extracts according to how these cells synthesize the extracellular matrix [33]. The biological mechanisms involved in the aging of dermal cells are key areas to understand skin aging. Large numbers of biological mechanisms, such as decreasing protein synthesis in the extracellular matrix or increasing degradation, are known to be altered during the skin aging process. In this context, as part of the ongoing search for ways to prevent and attenuate the skin aging mechanism, candidate compounds can be investigated in fibroblast culture models. Therefore, in this study, fibroblast cells were used to investigate skin anti-aging mechanisms including cell proliferation and antioxidant (enhancing SOD activity and nitric oxide inhibition) and collagenase inhibition through in vitro assays.

### 3.3.1. HDF Cell Proliferation

The ability of compounds to enhance the proliferation of HDF cells can be used to establish their wound healing and anti-aging effect to rejuvenate skin cells [34]. Older skin fibroblasts tend to migrate more slowly compared to younger cells, and promoting fibroblast proliferation is related to anti-aging properties. All samples promoted HDF cell proliferation in a dose-dependent manner (Figure 3A,B). At maximum nontoxic concentration, chlorogenic acid showed significantly higher proliferation-promoting activity at $18.46 \pm 1.01\%$ than caffeine and CBE at $16.22 \pm 0.86\%$ and $14.97 \pm 0.81\%$, respectively (Figure 4A). CBE showed enhancement of fibroblast cell proliferation in relation to the report by Affonso et al. that a hydrogel containing aqueous extract of green coffee showed good results in wound reduction (78.20%) [12]. Chen et al. reported that using 1% chlorogenic acid for a topical applied to Wistar rats for 15 days showed wound healing capacity with increased collagen synthesis via its antioxidant properties [35].

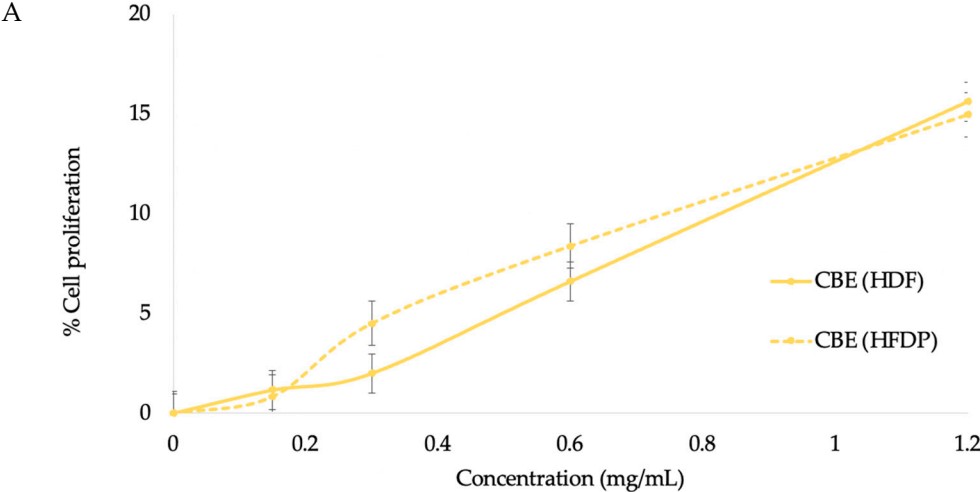

**Figure 3.** *Cont*.

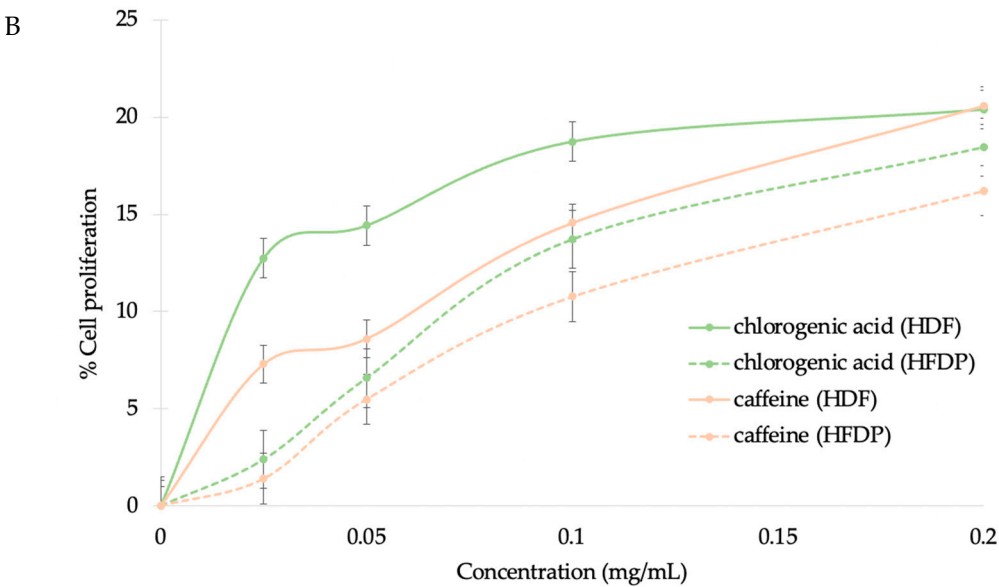

**Figure 3.** HDF and HFDP cell proliferation effect of various concentrations of CBE (**A**), caffeine, and chlorogenic acid (**B**).

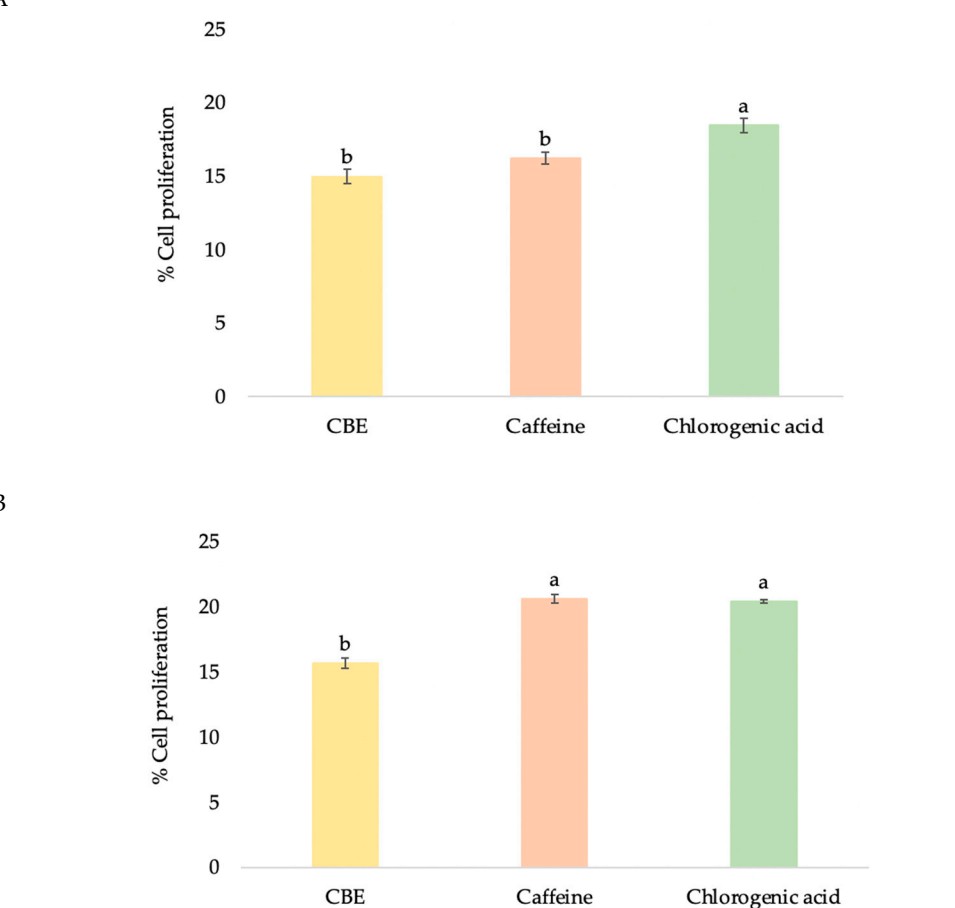

**Figure 4.** Effects of CBE, caffeine, and chlorogenic acid at maximum nontoxic concentrations on proliferation of HDF (**A**) and HFDP (**B**) cells. Data with different lower-case letters (a and b) indicate significant differences ($p < 0.05$) between samples.

### 3.3.2. Antioxidant Activities

SOD is a primary antioxidant enzyme that works against the toxic effect of superoxide radicals ($O_2-$) on cells by catalyzing hydrogen peroxide ($H_2O_2$) and oxygen ($O_2$) [36]. An imbalance between free radical production and antioxidant levels leads to oxidative stress, which was observed in the reduced activity of SOD antioxidant enzymes [37]. SOD is also a major antioxidant in human fibroblasts and is a causative factor in oxidative stress on telomere shortening [38]. Moreover, promotion of SOD activity can enhance the prevention of oxidative stress and photo-induced aging of the skin.

In order to investigate the SOD antioxidant-enhancing activity of CBE, a SOD determination kit that allows for a convenient assay by producing a water-soluble formazan dye upon reduction with a superoxide anion was used. The rate of reduction with $O_2$ is linearly related to the xanthine oxidase activity, which is inhibited by SOD. Therefore, SOD activity can be quantified by measuring the decrease in the color development of WST-1 formazan. The treatment with CBE induced significantly higher SOD activity ($74.53 \pm 0.92\%$) than that of caffeine ($12.56 \pm 2.43\%$), but lower activity than that of chlorogenic acid ($91.00 \pm 1.11\%$) (Figure 5A). These outcomes are in agreement with those reported by Zofia, who noticed that green coffee extract ($100\ \mu g/mL$) showed high activity of the superoxide dismutase enzyme, and its fermentation duration also had an effect on this ability, with 14-day fermented extract showing better results than extracts fermented for 7 and 28 days [14]. In addition, improvement of SOD and glutathione reductase (GR) activities by chlorogenic acid, caffeine and trigonelline were found in chronic coffee and caffeine ingestion mice [3].

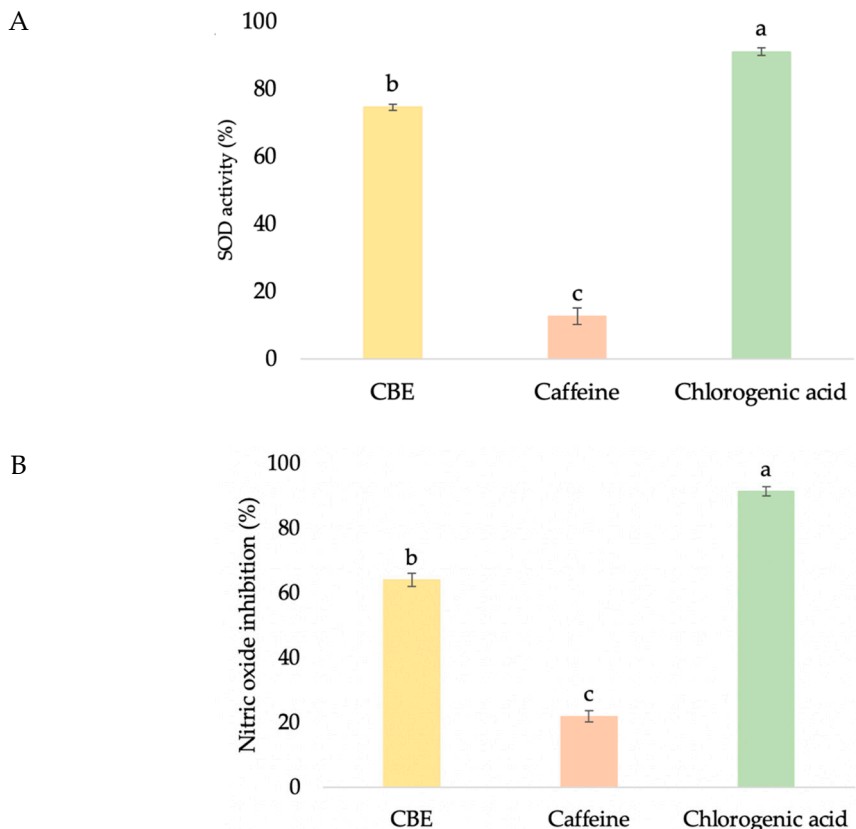

**Figure 5.** *Cont.*

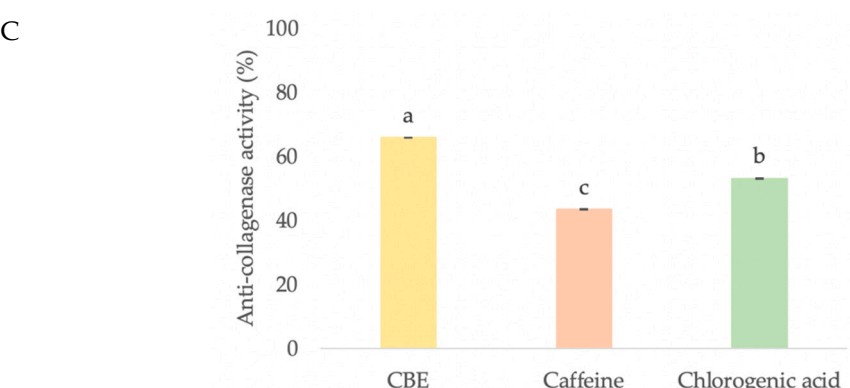

**Figure 5.** The superoxide dismutase (**A**), nitric oxide inhibition (**B**), and anti-collagenase (**C**) activities of CBE, caffeine, and chlorogenic acid on HDF. Data with different lower-case letters (a, b and c) indicate significant differences ($p < 0.05$) between samples.

Nitric oxide is involved in various types of inflammatory disorders by acting as an anti-inflammatory agent under normal physiological conditions. The overproduction of nitric oxide is considered to be a pro-inflammatory mediator and induces inflammation under abnormal conditions [39]. Moreover, nitric oxide is a short-lived radical and an important immune mediator with a profound cytotoxic and wound healing effect [40]. Thus, the determination of nitric oxide levels is a good indicator of cell inflammation. In this study, the amount of nitric oxide production in the culture medium was quantitated using Griess reagent after treating HDF with the test samples. Chlorogenic acid showed significantly higher inhibition of nitric oxide, by $91.29 \pm 1.41\%$ (Figure 5B). Although CBE ($63.93 \pm 2.05\%$) provided significantly lower inhibition activity than chlorogenic acid, it provided significant higher activity than caffeine ($21.81 \pm 1.78\%$) ($p < 0.05$). Lemos also reported that coffee bean exhibited inhibition of nitric oxide production in macrophage cells (RAW 264.7) [41]. However, 50 ug/mL of chlorogenic acid and caffeine revealed no significant nitric oxide inhibition [41], which was different from our result. This may be because their study used four-fold lower concentrations of pure compounds than our experiment.

Chlorogenic acid, which is a major phenolic compound in coffee, showed significantly higher antioxidant activities than alkaloids (caffeine) in both antioxidant assays ($p < 0.05$). Generally, many antioxidant experiments have proved that phenolic compounds are good free radical scavengers [42]. Many previous studies demonstrated that the extracts of coffees obtained from different fruit parts or locations and polarities of extraction solvent exhibited differences in antioxidant activities, and the literature makes it clear that the antioxidant activity depends mainly upon the presence of phenolic compounds. The interaction mechanism of the phenolic compounds has been correlated to the numbers and locations of functional groups, especially hydroxy groups, in the phenolic molecules, which directly affects their biological effects, with higher hydroxy groups resulting in an increase in activity and hydroxy groups in ortho position showing higher activity than those in meta- and para- positions. These results also strongly indicated that phenolic compounds in CBE are major contributors to their antioxidant capacity, which correlates with skin anti-aging capacity. Moreover, since the chlorogenic acid concentration in the CBE sample was about 30 ug/mL it can be assumed that the high antioxidant activities of CBE are due to the synergism effect of other antioxidants (epicatechin, catechin, rutin, protocatechuic acid, and ferulic acid, etc.) present in this extract.

### 3.3.3. Anti-Collagenase Activity

The clavation of interstitial collagens and alteration of the extracellular matrix are primarily responsible for the clinical manifestations of skin aging such as wrinkles, sagging, and laxity [43,44]. Hence, inhibition of collagenase was suggested as a key factor to slow down not only the loss of skin elasticity but also the progress of sagging [45].

Therefore, the evaluation of collagenase inhibition properties is the most often used assay to investigate the anti-aging effects of many natural products to be applied as cosmeceutical ingredients. To investigate the collagenase inhibition activity of CBE and reference compounds (caffeine and chlorogenic acid), bioassays were performed with a member of matrix metalloproteinases, MMP-1. Caffeine showed collagenase inhibition of 43.47%. The results corresponded to the report by Lee et al. [46], who established the inhibition of collagenase in a concentration-dependent manner of caffeine in an in silico experiment and an in vitro approach. Interestingly, CBE showed the highest significant inhibition of MMP-1, of 65.88%, followed by chlorogenic acid (53.19%) and caffeine (43.47%) (Figure 5C). This may be the result of the synergistic action of the chemically complex matrices of CBE. The results correspond with the study by McDaniel in which coffee berry extract showed down-regulated MMP-1 gene expression in biopsy skin samples that were taken and sectioned from participants who twice-daily applied CBE 1% cream and CBE 0.1% cleanser for 6 weeks [17].

### *3.4. Anti-Hair Loss Effect*

Nowadays, the number of persons who suffer from hair loss is increasing. Thus, it is quite relevant to develop cosmetic products to stop hair loss and enhance hair growth. HFDP cells were used to investigate the potential of new substances as novel therapeutic and cosmetic hair growth stimulating agents in hair care products. Dermal papilla cells are specialized mesenchymal cells located at the base of hair follicles that play essential roles in hair follicular morphogenesis and postnatal hair growth cycles. Therefore, these cells have been used to examine anti-hair loss and hair growth modulating effects of a variety of agents, natural extracts, growth factors, cytokines, peptides, and hormones. In addition, effects on a wide range of biomolecules and mechanistic pathways that play a key role in the biology of hair growth were also investigated using HFDP cells [47]. Several studies have shown that dermal papilla proliferation is closely associated with the hair growth cycle and an increase in dermal papilla cells in the anagen phase, causing a slowdown in hair loss. For the study reported herein, the test samples were investigated for their effects on cell proliferation, 5$\alpha$-reductase and the growth factor of HFDP cells.

### 3.4.1. HFDP Cell Proliferation

HFDP cell proliferation has been widely tested to determine the hair growth-promoting or hair-loss inhibiting effects from natural extracts [48]. HFDP cell proliferation prolonged the hair cycle's anagen phase, which increased hair density and reduced hair loss [36]. In this study, the mitogenic effect of CBE was investigated on the dermal papilla cell culture system. All samples promoted the proliferation of HFDP cells in a dose-dependent manner, with 1.2 mg/mL of CBE stimulating hair cell proliferation at 15.64 $\pm$ 1.59% (Figure 3A,B), whereas 0.2 mg/mL of caffeine (20.60 $\pm$ 0.63%) and chlorogenic acid (20.40 $\pm$ 1.54%) showed significantly higher promotion of HFDP cell proliferation than CBE (Figure 4B). Enhancing HFDP cell proliferation with the test sample showed its possibility to extend the anagen phase, which is related to healthy hair and reduction of hair loss. In a previous study, Kim et al. reported that after being treated with 10, 20 and 50 ppm of caffeine, human dermal papilla cells showed increased proliferation of 10, 18, and 19%, respectively, in a CCK-8 assay, which was related to hair growth promoting results [49].

### 3.4.2. 5$\alpha$-Reductase Inhibition

Not only cell proliferation but also androgens are involved in hair growth. Testosterone and dihydrotestosterone are two major androgens that indirectly control hair growth. Hair loss is induced by dihydrotestosterone, which is converted from testosterone by 5$\alpha$-reductase. Increasing the androgenic signal decreases the duration of the anagen phase [50,51], and the response to the transformation of large follicles to smaller ones induces progressive thinning of the scalp hair and hair loss [50,52]. The inhibition of 5$\alpha$-

reductase activity in dermal papilla cells should have a direct effect on androgenic alopecia treatment [22].

Previous studies have determined that caffeine inhibits the activity of the 5α-reductase enzyme and contributes to a renewed growth phase for human hair. Thus, caffeine was identified as a stimulator of human hair growth *in vitro*, which may have an important clinical impact on managing androgenetic alopecia [53]. Lademann et al. [54] and Teichmann et al. [55] demonstrated that shampoo containing caffeine efficiently penetrated into the hair follicles of the human scalp and affected hair growth. Thus, caffeine is used in hair care products that are claimed to reduce and slow down balding and also stimulate hair growth. In our previous study, in order to obtain caffeine as a major component in coffee berry extract, we extracted the coffee fruit with ethanol and investigated its 5α-reductase inhibition activity in human prostate cancer cell lines (DU−145). The extract showed a 16.5% reduction in 5α-reductase activity. In a clinical trial with 13 men and 34 women, daily application of hair tonic spray containing 10% extract for 3 months showed significant reduction in hair loss numbers after the combing test [56]. In this study, we extracted the sample with higher polarity solvent (50% ethanol) to obtain higher phenolic compounds [16] and evaluated the inhibition of 5α-reductase activity in HFDP cells. When the cells were incubated with the samples at various concentrations, there was a dose-dependent 5α-reductase inhibition (Figure 6). Surprisingly, at the maximum nontoxic concentration, CBE (67.59 ± 2.43%) and chlorogenic acid (66.35 ± 1.30%) showed higher inhibition of 5α-reductase activity than the known inhibitor, caffeine (55.87 ± 1.04%) (Figure 7). Based on our best knowledge, this is the first report on the comparison of 5α-reductase inhibition activity of caffeine, chlorogenic acid and coffee berry extract. These results suggested that the hair growth-promoting effect of CBE mediated through the 5α-reductase inhibition pathway was due to its chlorogenic acid and caffeine content, which provided a synergistic effect.

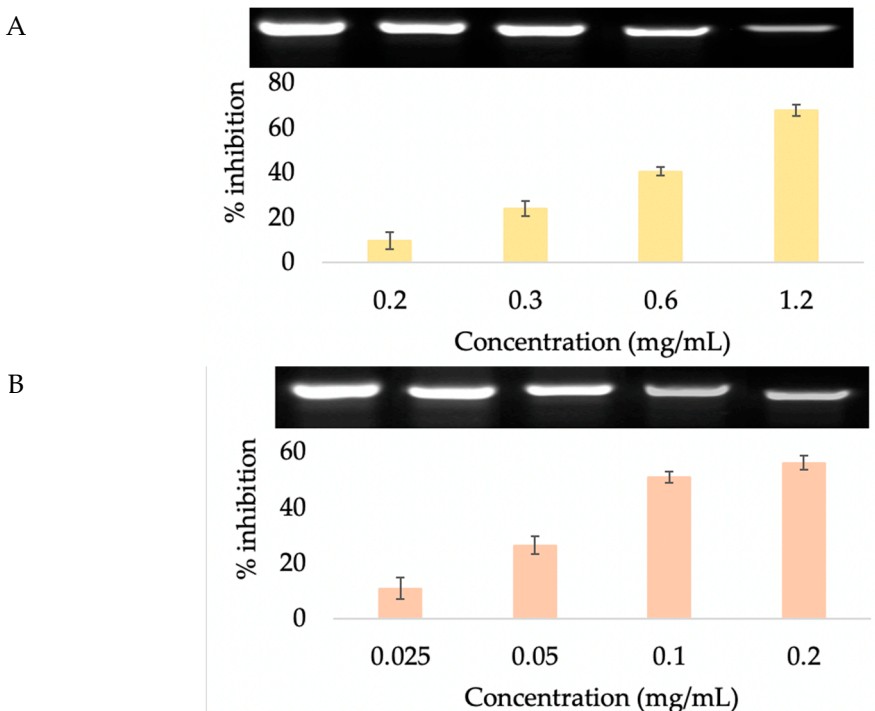

**Figure 6.** *Cont.*

C

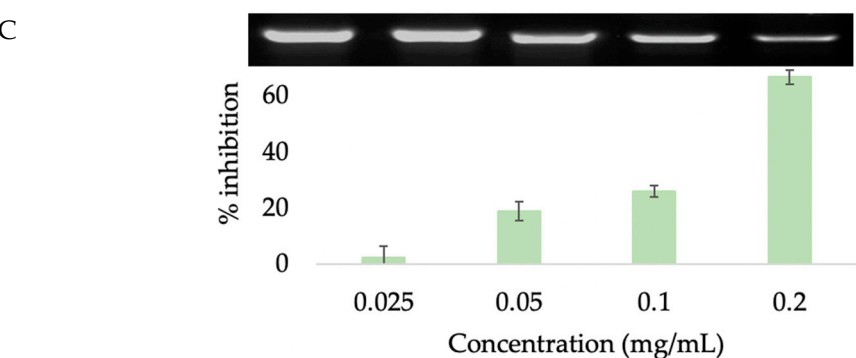

**Figure 6.** 5α-reductase inhibition of (**A**) CBE, (**B**) caffeine, and (**C**) chlorogenic acid on HFDP cells.

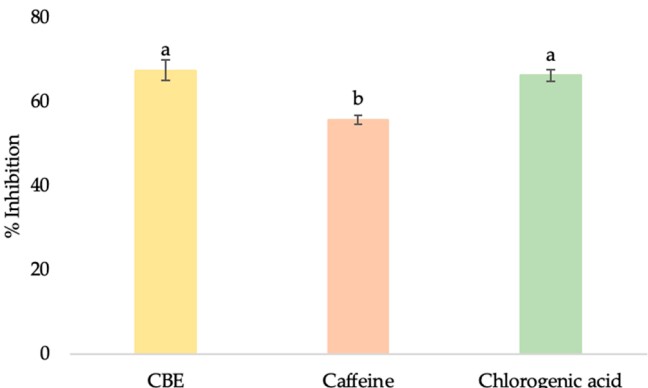

**Figure 7.** 5α-reductase inhibition of CBE, caffeine, and chlorogenic acid at maximum nontoxic concentration. Data with different lower-case letters (a and b) indicate significant differences ($p < 0.05$) between samples.

### 3.4.3. Growth Factor Gene Expression

The gene expression level of growth factor is one method to elucidate the mechanism of compounds in hair growth-promoting activity. There are several growth factors that are implicated in the regulation of hair growth, such as IGF-1, KGF, HGF and VEGF. The expression of several growth factors is involved in the regulation of hair morphogenesis and hair growth [57]. IGF-1 stimulates the growth of epithelial cells and regulates cellular proliferation and migration during hair follicle development [58,59]. IGF-1 promotes hair growth and plays an important role in follicular proliferation, tissue remodeling, and the hair growth cycle, and the decreased secretion of IGF-1 was correlated with androgenetic alopecia [60,61]. KGF is an endogenous mediator of hair follicle growth and stimulates sebaceous glands and hair follicle keratinocytes' development [62]. Furthermore, KGF is one of the growth factors that stimulates hair fiber elongation, protects hair follicles from cell death, and directly affects the development of hair follicles [63]. HGF is a paracrine factor that promotes follicular growth via action on neighboring follicular epithelial cells [64]. HGF has been shown to be secreted by dermal papilla cells that are utilized to manipulate the sheath fibroblasts surrounding hair follicles and stimulate hair regeneration [65]. VEGF is an important mediator of angiogenesis in hair follicles by improving perifollicular vascularization and increasing hair follicle size [66–68]. The importance of VEGF in hair growth stimulation has been further supported by a study on the activity of minoxidil (a well-known drug for androgenic alopecia) [69]. To clarify the molecular mechanism of compounds in hair growth-promoting activity, these growth factors were determined. In this study, the expression levels of these growth factors were measured in HFDP cells when treated with and without CBE, caffeine, and chlorogenic acid. As compared to the untreated group (1.00-fold), the results showed that HFDP cells treated with caffeine (2.04-fold) showed significantly higher levels of IGF-1 expression than cells treated with CBE (1.69-fold) and chlorogenic acid (0.84-fold) (Figure 8A). Caffeine (1.85-fold) and CBE

(1.58-fold) showed significantly higher levels of KGF expression than chlorogenic acid (1.30-fold) (Figure 8B). However, there was not a significant difference in the expression of the HGF gene between caffeine (0.44-fold) and CBE (0.28-fold) or caffeine and chlorogenic acid (0.54-fold)), and there was also lower expression of HGF than of other growth factors in this study (Figure 8C). Caffeine showed significantly higher induced VEGF expression (2.33-fold) than CBE (1.74-fold) and chlorogenic acid (0.63-fold) (Figure 8D).

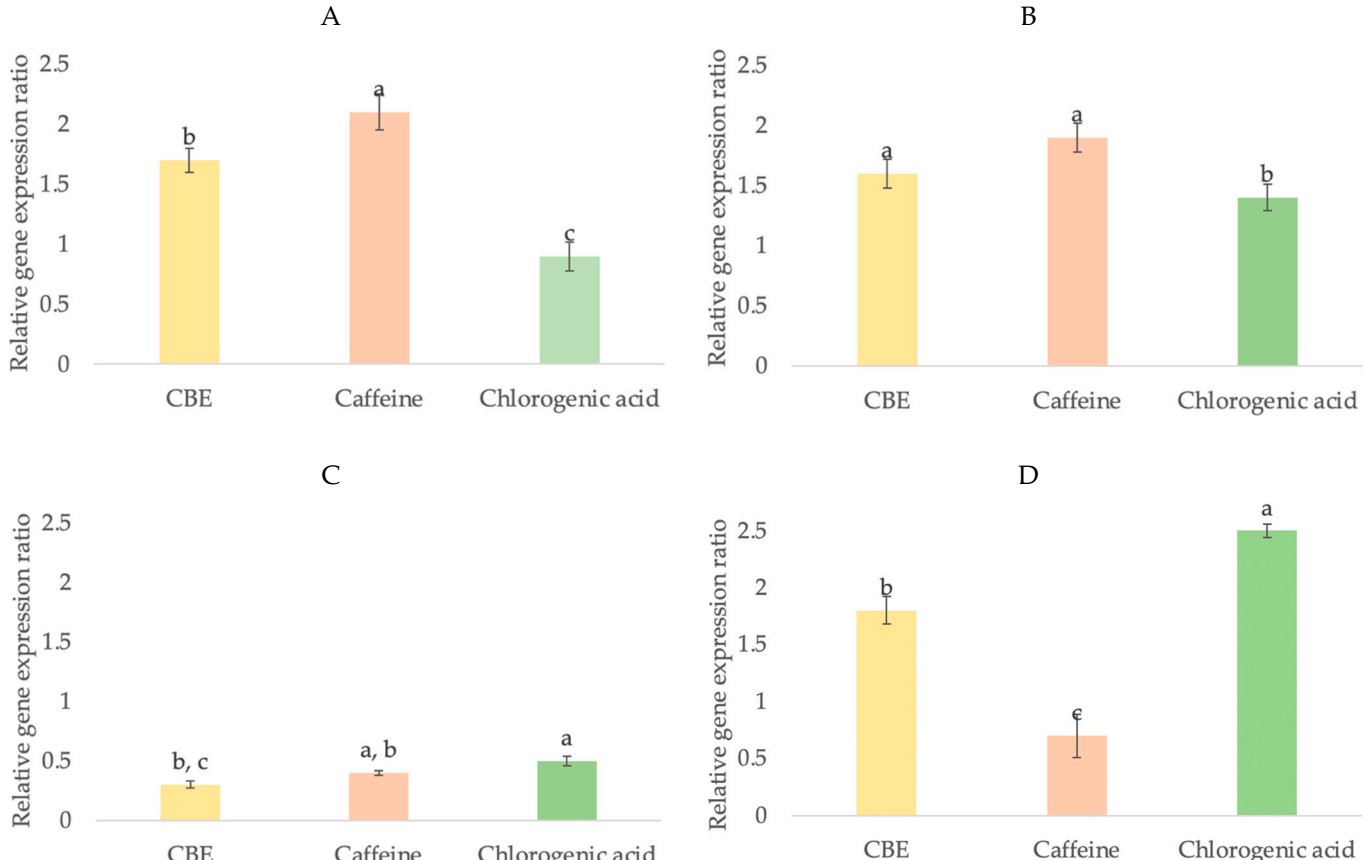

**Figure 8.** Relative fold expression levels of IGF-1 (**A**), KGF (**B**), HGF (**C**) and VEGF (**D**) genes in HFDP cells. Values are presented as means $\pm$ SD ($n = 3$). Data with different lower-case letters (a, b and c) indicate significant differences ($p < 0.05$) between samples.

The results showed that caffeine and chlorogenic acid influence hair growth via different pathways. In this study, caffeine showed a high effect on IGF-1, KGF and VEGF genes, which indicated its effect on epithelial cells and hair follicle keratinocytes' development together with a strong effect on the induction of vasodilation of scalp blood vessels. These results are consistent with the report by Fischer et al. that caffeine enhanced hair shaft elongation, prolonged anagen duration, stimulated hair matrix keratinocyte proliferation and upregulated IGF-1 gene expression [70]. Kim et al. also reported that after treatment with caffeine (20 ppm) and minoxidil (10 ppm, as positive control), the mRNA level in human dermal papilla cells increased in KGF by 1.6 and 1.7-fold, respectively, higher than in the untreated group and caffeine increased VEGF gene expression more than minoxidil [49]. Chlorogenic acid increased KGF expression compared to untreated cells, implying that it has an impact on hair follicle development. However, any of the samples have no effect on HGF, which is related to follicular growth and encourages hair regeneration.

In this regard, it is noteworthy that CBE has broader potential to increase hair growth mRNA expression, especially of IGF-1, KGF, and VEGF, in comparison with the pure compound, chlorogenic acid, due to CBE containing both caffeine and chlorogenic acid components. The results indicated that CBE could stimulate hair growth by promoting

epithelial cell proliferation around the base of hair follicles, hair follicle keratinocyte development, and vasodilation of scalp blood vessels.

## 4. Conclusions

The major compounds of CBE were identified as caffeine and chlorogenic acid by HPLC. CBE showed skin anti-aging properties including enhanced keratinocyte cell proliferation and SOD activity and inhibited nitric oxide and collagenase activities. Moreover, CBE showed an increase in dermal papilla cell numbers as well as the mRNA expression of hair growth-related factors including IGF-1, KGF and VEGF, the essential growth factors for prolonging the anagen (growth) phase in the hair cycle. CBE also exhibited good inhibitory activity against 5$\alpha$-reductase. Therefore, our findings confirmed that CBE has high potential to be used as an active ingredient for skin anti-aging, anti-hair loss and hair growth stimulation in not only skin care but also hair care product applications.

To our knowledge, the present work is the first report comparing anti-hair loss and hair growth promoting effects of caffeine, chlorogenic acid and coffee berry extract in cell based assays.

**Funding:** This work was financially supported by the Office of National Higher Education Science Research and Innovation Policy Council through Program Management Unit for Competitiveness, grant number C10F640029.

**Institutional Review Board Statement:** Not applicable.

**Informed Consent Statement:** Not applicable.

**Data Availability Statement:** All of the data are available in the manuscript.

**Acknowledgments:** The author expresses thanks to the Office of National Higher Education Science Research and Innovation Policy Council through Program Management Unit for Competitiveness for research funding and Cosmetic and Beauty Innovations for Sustainable Development (CBIS) Research Group and Scientific and Technological Instruments Center, Mae Fah Luang University for providing scientific equipment and facilities for this work.

**Conflicts of Interest:** The author declares no conflict of interest.

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
