# Peer review of "Effect of Coffee Berry Extract on Anti-Aging for Skin and Hair—In Vitro Approach"

_cosmetics, doi:10.3390/cosmetics9030066_

Round 1

Reviewer 1 Report

The author investigated the antioxidant and anti-aging effects of coffee berry extract (CBE) in human dermal fibroblast (HDF) and hair follicle dermal papilla (HFDP) cells. The antioxidant effect of CBE was studied using caffeine and chlorogenic acid as a comparing group, but since CBE contains both caffeine and chlorogenic acid, it is expected that CBE exhibits the similar effect of treatment with both caffeine and chlorogenic acid.

I'm rather curious about if it is an unknown single molecule component of CBE, which shows less toxic than caffeine and chlorogenic acid, has antioxidant effects. It would be interesting if a new single molecule other than caffeine and chlorogenic acid showed the antioxidant effect of CBE.

Therefore, I think it would be interesting to investigate further whether CBE is composed of caffeine and chlorogenic acid at a certain content ratio or concentration and exhibited antioxidant effects while being non-toxic. The results of this study are noteworthy in that CBE has similar antioxidant effects in human dermal fibroblast (HDF) and hair follicle dermal papilla (HFDP) cells as caffeine and chlorogenic acid, which are already known. However, I think it would be a much better study if the author adds future research on the exact molecules or a certain content ratio of CBE that exerts these positive effects.

But overall, I think that the direct effect of CBE in human dermal fibroblast (HDF) and hair follicle dermal papilla (HFDP) cells is a very meaningful and has general interest study.

I have several concerns about this study. In the introduction part, it would be better to add the reports related to coffee and aging for better understanding of significance of this research on the anti-aging effect of coffee. The second is to determine the content and concentration of caffeine and chlorogenic acid in CBE. It is thought necessary to understand whether a complex compound of CBE compared to a single compound of each caffeine and chlorogenic acid further enhances the antioxidant effect while being non-toxic. Also, I think it is necessary to show the figure format in the same way, the clear explanations in the figure legend including statistical analysis in more detail for easy understanding of each figure.

Author Response

In the introduction part, it would be better to add thereports related to coffee and aging for better understanding of significance of this research on theanti-aging effect of coffee.

The reports related to coffee and aging is added more in introduction part.

The second is to determine the content and concentration of caffeine and chlorogenic acid in CBE. It is thought necessary to understand whether a complexcompound of CBE compared to a single compound of each caffeine and chlorogenic acid further enhances the antioxidant effect while being non-toxic.

The other compounds in CBE that may ne enhanced itsantioxidant ability is added in discussion part.

Also, I think it is necessary to show the figure format in the same way, the clear explanations in the figure legend including statistical analysis in more detail for easy understanding of each figure.

The figure format is changed in the same way.

The explanations in the figure legend including statistical analysis is added.

Reviewer 2 Report

The work presented to me for review, entitled " Effect of coffee berry extract on anti-aging for skin and hair, is an original work. The author try to explain skin anti-aging and anti-hair loss along with hair growth promoting potential of CBE on human skin (HDF) and hair (HFDP) cells. The presented studies are an interesting element of the potential use of coffee extract in cosmetology. Overall, the topic is interesting and presented in an interesting way. The abstract and the introduction are written correctly with all the key elements undertaken by the author, although I would suggest improving the purpose of the work a little more because it is too complex. The Materials and Methods section is a bit too general and requires more details to be added, for example the section on gene expression, please add conditions, primer sequences, and gene names in italics. Besides, the results and discussion section is not very informative, little discussed, although the author emphasizes that these are the first results, I think that the coffee extract itself is so known that there are probably works that can be discussed in terms of the compounds contained in the extract and similar studies. The enzymatic tests presented by the author are quite common, so it should not be a problem to discuss them in the context of even coffee extract. Please also remember that this type of research is just a prelude to describing the beneficial effects of this extract on patients, the more so as it can sometimes be quite allergenic. Conclusions are written correctly and summarize the results of the author's studies. The tables and figures presented by the author are adequate to the presented results. Please correct some punctuation and stylistic errors in the manuscript.

Author Response

The Materials and Methods section is a bit too general and requires more details to be added, for example the section on gene expression, please add conditions, primer sequences, and gene names in italics.

Materials and Methods section is added more details.

Besides, the results and discussion section is not very informative, little discussed, although the author emphasizes that these are the first results, I think that the coffee extract itself is so known that there are probably works that can be discussed in terms of the compounds contained in the extract and similar studies.

The relation of compounds contained in the extract and anti-aging is discussed in more detail.

The enzymatic tests presented by the author are quite common, so it should not be a problem to discuss them in the context of even coffee extract.

The enzymatic tests is more discussed.

Please also remember that this type of research is just a prelude to describing the beneficial effects of this extract on patients, the more so as it can sometimes be quite allergenic.

The safety detail of coffee is added.

The tables and figures presented by the author are adequate to the presented results. Please correct somepunctuation and stylistic errors in the manuscript.

Some punctuation and stylistic errors is corrected.

Title should be modified for "Effect of coffee berry extract on anti-aging for skin and hair - in vitroapproach" since it may lead to misunderstandings for readers. Just in vitro studies were performed.

Title is modified.

Reviewer 3 Report

The manuscript entitled “Effect of coffee berry extract on anti-aging for skin and hair” by Saewan intends to report the effects of coffee berry as cosmetic ingredient. The manuscript is very interesting and in the scopus of the journal. The English is good, and few mistakes and gramma errors can be found. My concerns:

  • Title should be modified for “Effect of coffee berry extract on anti-aging for skin and hair – in vitro approach” since it may lead to misunderstandings for readers. Just in vitro studies were performed.
  • More information regarding cell origin (ATCC??) should be included
  • Line 328-330 : revise English
  • Discussion should be improved and detailed, comparing the results with other studies that used coffee products as cosmetic ingredients.

Author Response

More information regarding cell origin (ATCC??) should be included

Cell origin is included.

Line 328-330 : revise English

Line 328-330 : English is revised.

Discussion should be improved and detailed, comparing the results with other studies that used coffee products as cosmetic ingredients.

Discussion is be improved in comparing the results with other studies.

Round 2

Reviewer 1 Report

I think the current version is much improved overall. However, the figure legend still lacks explanation. Fig. 5 is missing what percentage the y-axis represents. And in the Fig. 7, I still can not understand the meaning of statistical A, B, C, a, b, and c. It would be good if the detailed explanation of the figure legend could be showed.

Author Response

The figure 5 and 7 have been changed.

Reviewer 3 Report

The authors revised the manuscript according to the reviewers concerns. In my opinion the paper is in conditions to be accepted.

Author Response

I revised the manuscript as reviewer suggestion